# OBJECT TRACKING BY HIERARCHICAL PART-WHOLE ATTENTION

## ABSTRACT

We present in this paper that hierarchical representations of objects can provide an informative and low-noisy proxy to associate objects of interest in multi-object tracking. This is aligned with our intuition that we usually only need to compare a little region of the body of target objects to distinguish them from other objects. We build the hierarchical representation in levels of (1) target body parts, (2) the whole target body, and (3) the union area of the target and other objects of overlap. Furthermore, with the spatio-temporal attention mechanism by transformer, we can solve the tracking in a global fashion and keeps the process online. We design our method by combining the representation with the transformer and name it Hierarchical Part-Whole Attention, or HiPWA for short. The experiments on multiple datasets suggest its good effectiveness. Moreover, previous methods mostly focus on leveraging transformers to exploit long temporal context during association which requires heavy computation resources. But HiPWA focuses on a more informative representation of objects on every single frame instead. So it is more robust with the length of temporal context and more computationally economic.

## 1 INTRODUCTION

How to represent the visual existence of an object in a discriminative fashion is a core question of computer vision. In this paper, we propose a hierarchical part-whole representation to represent the visual existence of objects. We adopt multi-object tracking as the application area since the distinguishable appearance feature is critical to avoid the mismatch among target objects when tracking across frames. To gather and process the visual information from different levels, we combine the hierarchical part-whole representation with the attention mechanism from transformers to summarize distinguishable and discriminative visual representations for objects of interest.

In the task of multi-object tracking, given a bounding box to localize objects of interest, how should we recognize the major object within the box and distinguish it from the background and other objects, especially some also having partial existence in the box? We believe the visual specificity of one object comes from three perspectives: the *compositional*, the *semantic* and the *contextual*. The compositional suggests the salient and unique visual regions on an object, such as a hat on a pedestrian whose color is different from all others in the same image. With a salient visual composition attached to an object, we can track it across frames even without seeing its full body. The semantic visual information is the commonly adopted one in modern computer vision such as a tight bounding box or instance segmentation mask. It defines the occupancy area of the object with the bond between its visual existence and semantic concept. Finally, contextual visual information describes the surroundings of an object. It helps to distinguish an object via contrast. For example, the bounding box might contain pixels from the background and secondary objects. However, a tight bounding box offers a strong underlying prior when combined with visual context: an object whose parts span across the boundary of the bounding box should not be the major object of this bounding box. Being the secondary object or not an object of interest, it should be regarded as noise when we generate a distinguishable visual representation for the major subject in the bounding box. The analysis above shows each level has its value to represent an object discriminatively. Motivated by the insight, we propose to represent an object by a three-level hierarchy: body parts, full body, and the union area including objects with overlap. We summarize it as a "Part-Body-Union" hierarchy.

With the hierarchy constructed, an ideal path to solving the target association in multi-object tracking is to leverage the salient information within the body area and discard mismatch by eliminating the

noise revealed by the contextual contrast. Without requiring more fine-grained data annotation, we propose to use transformers to process the hierarchical representation as the attention mechanism can discover important visual information. So, by combining the hierarchical visual representation and attention-based feature fusion, we finally propose our method as Hierarchical Part-Whole Attention, or HiPWA for short. In this work, we build a baseline model following this design and demonstrate its effectiveness in solving multi-object tracking problems. Through experiments on multiple multi-object tracking datasets, the proposed method achieves comparable or even better performance than the state-of-the-art transformer-based methods with a more lightweight implementation and better time efficiency during training and inference.

## 2 RELATED WORKS

### 2.1 REPRESENTING OBJECTS BY PARTS

The most commonly used object representation for multi-object tracking is bounding boxes. However, the bounding box is noisy by containing background pixels and pixels from secondary objects. On the other hand, our life experience demonstrates that, in many scenarios, it is not necessary to observe the full body of objects to specify an object visually and tracking targets by the distinguishable parts on it is usually more efficient. Therefore, researchers also have been studying object detection and tracking with more fine-grained representation. A common way is to use pre-defined certain parts on target bodies, such as only human head (Sundararaman et al., 2021; Shao et al., 2018), human joints (Andriluka et al., 2018; Xiu et al., 2018) or even every pixel (Voigtlaender et al., 2019; Weber et al., 2021). However, all these choices require more fine-grained data annotation beyond bounding boxes and more fine-grained perception modules beyond just normally available object detectors. In the contrast, the part-whole hierarchy we construct requires no additional annotations and we still solve tracking tasks at the granularity of bounding boxes. The idea of modeling objects with different levels is inspired by the hierarchical modeling of the human body (Marr, 2010) by David Marr when he explains how to construct the visual structure of an object from primal sketch to 2.5 sketch and further 3D representation. His classic three levels of visual information processing system concludes this in a higher-level: the *computational*, the *algorithmic*, and the *implementational*. A similar theory is also introduced by Fodor & Pylyshyn (1988) as the *semantic*, the *syntactic*, and the *physical*. Compared to these cognitive theories aiming to model general visual representation, the three perspectives we propose to recognize an object and distinguish it from others (the *compositional*, the *semantic* and the *contextual*) only apply to the specific problem of generating an effective visual descriptor to represent the objects of interest.

### 2.2 TRANSFORMER-BASED MULTI-OBJECT TRACKING

Transformer (Vaswani et al., 2017) is originally proposed for natural language processing. It shows a powerful capacity for information representation and processing. Later, DETR (Carion et al., 2020) introduces the transformer to the area of visual perception for object detection. It models object detection as solving a bipartite matching problem. Given that the matching-based strategy by DETR is quite similar to the target matching in the task of multi-object tracking, it is intuitive to further migrate transformer to this area. TransTrack (Sun et al., 2020) is the first work using the transformer to solve the MOT problem but it does not invent any association strategy by transformers. A concurrent work TrackFormer (Meinhardt et al., 2021) takes a further step to use the cross attention in transformer decoder in the stage of association by query passing. On the other hand, VisTR (Wang et al., 2021c) proposes a novel global association scheme upon transformer where a video clip of multiple frames is forward into the transformer at the same time to associate objects within the clip. More recently, many works (Zhou et al., 2022; Zeng et al., 2021) follow the global association scheme in either training or inference and achieve good performance. A key to their success is to process the information over a long temporal period, which can be hardly handled without the transformer. GTR (Zhou et al., 2022) makes a baseline model of using only appearance in associating objects and removing some secondary modules such as positional encoding and learnable object query. However, a downside of processing multiple frames as a batch by the transformer is the high requirement of computation resources. It has become a common practice to train the model on at least 4xV100 GPUs (Zhou et al., 2022; Sun et al., 2020; Zeng et al., 2021) or even 8xA100 GPUs (Cai et al., 2022). These methods usually suffer from significant performance drop if only limited computation resource is available. This is because they usually make improvements to association performance by taking advantage of a long temporal window and gathering more visual context within it. In

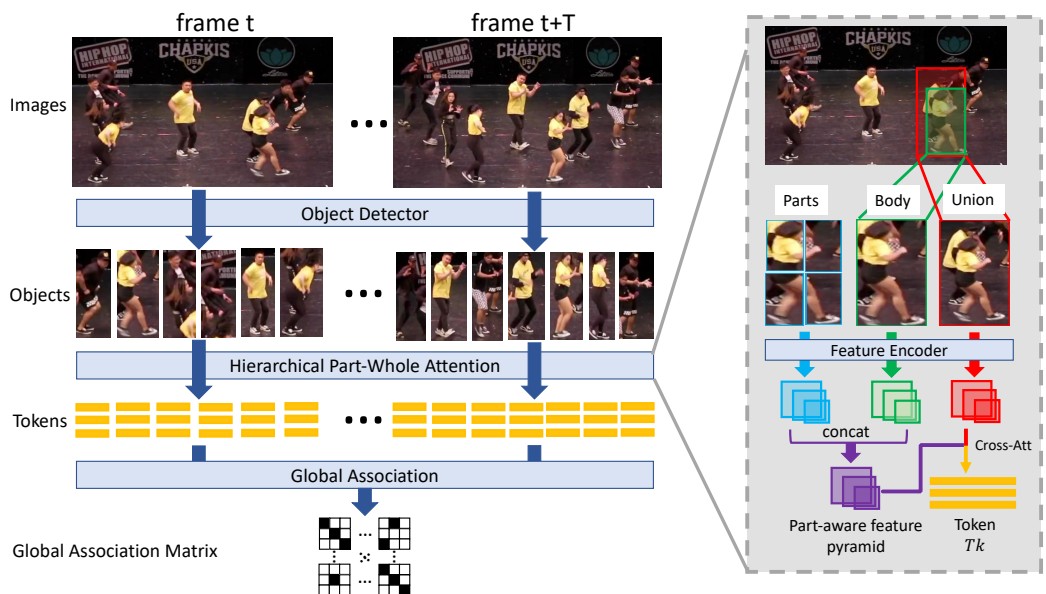

**Figure 1:** The pipeline of our proposed method. Our method follows the tracking-by-detection scheme and conducts association in a global fashion. Our proposed hierarchical feature attention module fuses the features from three levels, i.e., object parts (*compositional*), object bodies (*semantic*) and union area of objects with overlap (*contextual*). The output features serve as tokens in the following transformer decoder for the global association.

this work, we focus on building a more computation and memory-efficient visual representation for objects from the scope of a single frame instead. This scheme is flexible to be integrated with transformers and more robust to short time windows during object association.

## 3 METHOD

In this section, we introduce the method we propose to leverage a hierarchical part-whole visual representation with the attention mechanism from the transformer for multi-object tracking. In Section 3.1, we describe the overview structure of our method using global association. Then, in Section 3.2, we dive into the details of our proposed part-whole attention module. Finally, we talk about the details of training and inference by HiPWA in Section 3.3.

### 3.1 GLOBAL ASSOCIATION

Before the transformer is introduced into this area, people usually solve multi-object tracking in a frame-by-frame fashion where the association is performed on only two frames. Recently, the transformer shows the advantage to gather and process information from multiple steps in parallel. To leverage this advantage, previous methods (Wang et al., 2021c; Zhou et al., 2022) propose to perform association in a video clip instead of just two frames. Therefore, the spatio-temporal attention capacity of the transformer leads to a new global association fashion. We follow this scheme in our proposed method. The overall pipeline of HiPWA is shown in the left-hand half of Figure 1. Now, we explain the three stages of it.

**Detection and Feature Extraction.** Given a video clip of $T$ frames, i.e., $\mathcal{T} = \{t, t+1, ..., t+T\}$, we have the corresponding images $\mathcal{I} = \{I^t, I^{t+1}, ..., I^{t+T}\}$. Given a detector model, we could derive the detections of the object category of interest on all frames in parallel, noted as $\mathcal{O} = \{O_1^{t_1}, ..., O_N^{t_N}\}$. $N$ is the number of detections on the $T$ frames and $t_i \in \mathcal{T}$ ($1 \le i \le N$) is the time step where the $i$-th detection, i.e., $O_i^{t_i}$, is detected. Then, we generate the representations of each detected object and note them as $\mathcal{F} = \{F_1, F_2, ..., F_N\}$. The most commonly adopted solution is to use the backbone output on the object area as the representation features while we adopt our proposed hierarchical part-whole representation instead whose details are to be introduced soon.

**Token Generation by Hierarchical Part-Whole Attention.** After being projected into vectors, the hierarchical representations of detections become tokens $\mathbf{T}^{\text{det}} = \{Tk_1^{\text{det}}, Tk_2^{\text{det}}, ..., Tk_N^{\text{det}}\}$,

which are also terms as "object query" in previous works (Sun et al., 2020; Meinhardt et al., 2021). Concatenating the tokens makes $Q^{\text{det}} \in \mathbb{R}^{N \times D}$, where $D$ is the feature dimension. If we aim to associate the new-coming detections with existing trajectories, we also need the tokens to represent the existing $M$ trajectories, i.e., $\mathbf{T}^{\text{traj}} = \{Tk_1^{\text{traj}}, Tk_2^{\text{traj}}, ..., Tk_M^{\text{traj}}\}$. The transformer has shown good power to generate more discriminative feature tokens for trajectories by iterative query passing (Zeng et al., 2021) or long-time feature buffering (Cai et al., 2022). But to make our method simple, we directly project the hierarchical representation of objects on existing trajectories to represent the trajectories. Given a historical horizon $H$ to backtrack the objects on the previous time steps of a trajectory, we represent a trajectory, $Tk_j^{\text{traj}}$, with the "track query" $Q_j^{\text{traj}} \in \mathbb{R}^{H \times D}$. The track query is the combination of the feature tokens of detections within the historical horizon on the corresponding trajectory.

**Global Association.** By cross-attention, we could get the association score between the set of detections and the trajectory $Tk_j^{\text{traj}}$ as $S(Q_j^{\text{traj}}, Q^{\text{det}}) \in \mathbb{R}^{H \times N}$. In practice, because we aim to associate between all $M$ trajectories and $N$ detections, we perform the cross-attention on all object queries and track queries at the same time, namely $S(Q^{\text{traj}}, Q^{\text{det}}) \in \mathbb{R}^{HM \times N}$. By averaging the score on the $H$ frames selected from the historical horizon, we could finally get the association score between detections and trajectories as $\mathbf{S} \in \mathbb{R}^{M \times N}$. Then, we need to make sure that a trajectory will never be associated with more than one object from the same frame. We normalize the association scores between a trajectory and objects from the same time step by softmax. So the normalized association score between the $j$-th trajectory and the $i$-th detection is

$$P(\mathbf{M}_{j,i}^{\text{asso}} = 1|Q^{\text{det}}, Q^{\text{traj}}) = \frac{\exp(\mathbf{S}_{j,i})}{\sum_{k \in \{1,2,...,N\}} \mathbf{1}[t_k = t_i]\exp(\mathbf{S}_{j,k})}, \tag{1}$$

where the binary indicator function $\mathbf{1}[t_k = t_i]$ indicates whether the $i$-th detection and the $k$-th detection are on the same time step. $\mathbf{M}^{\text{asso}} \in \mathbb{R}^{(M+1) \times N}$ is the final global association matrix. Its dimension is of $(M + 1) \times N$ because each detection can be associated with an "empty trajectory" to start a new track in practice. The query of the "empty trajectory" is represented by a query randomly drawn from previous unassociated tokens during training. Also, after the association, unassociated trajectories will be considered absent on the corresponding frames. In such a fashion, we can train over a large set of detections and trajectories in parallel and also conduct inference in an online manner by setting $\mathcal{O}$ as the set of the detections only from the new-coming frame.

### 3.2 HIERARCHICAL PART-WHOLE ATTENTION

Finally, we come to the details about constructing hierarchical part-whole visual representations. We name this process hierarchical part-whole attention as we use the attention mechanism to gather and process information from different levels in the hierarchy, which is illustrated in the right-hand half of Figure 1. We design this representation because we think there are three perspectives to describe the existence of an object and its identification over other objects: the *compositional*, the *semantic*, and the *contextual*. Correspondingly, we think the body part patches, the full object body, and the union of the occupancy of objects with interaction provide the knowledge from the three perspectives respectively. The insight behind this module is what we would like the most to deliver in this work.

**Hierarchy Construction.** We represent a detected object by a quintuple, i.e., $O = [x, y, w, h, c]$, where the first four values describe its bounding box and $c$ is the detection confidence. So its body area is $B = [x, y, x + w, y + h]$. Next, we divide the body into multiple sub-regions (parts). By default, similar to what ViT (Dosovitskiy et al., 2020) does upon images, we divide the bounding boxes into $2 \times 2$ bins, making a set of body parts as $\mathcal{P} = \{P_1, P_2, P_3, P_4\}$. On the other hand, from a global scope, there are other targets interacting with $O$ which are highly likely to be mismatched with $O$ in the association stage. We crop the union area enclosing $O$ and all other targets having overlap with it. We note the union area as $U$. Till now, we have derives the part-whole hierarchy $\{\mathcal{P}, B, U\}$ in a very straightforward way.

**Feature Fusion.** Given the part-whole hierarchy, we have to fuse the features from different levels to get the final feature tokens for association. With a feature encoder, we can extract the CNN features from them as $F_{\mathcal{P}} \in \mathbb{R}^{4C \times H \times W}$, $F_B \in \mathbb{R}^{C \times H \times W}$ and $F_U \in \mathbb{R}^{C \times H \times W}$. We simply concatenate the features from the first two levels as $F_{\text{P+B}} \in \mathbb{R}^{5C \times H \times W}$. Then, by a two-layer

projection network, we gain projected features $V_{P+B} \in \mathbb{R}^{5 \times D}$. We also apply the projection to the union area features and get $V_U \in \mathbb{R}^D$. Finally, we perform cross-attention between $V_{P+B}$ and $V_U$ and forward the output to an MLP network to get the tokens of shape $\mathbb{R}^{5 \times D}$. Before being forwarded to the global association stage, the tokens would be projected to the uniform dimension of $D$.

## 3.3 TRAINING AND INFERENCE

The method we implement is a baseline model without complicated designs on "queries". We simply use the hierarchical part-whole features of detected objects to serve as the representations of both detections and trajectories. And during training, we can associate between detections in the sampled video clips or between detections and existing trajectories. These two schemes of associations thus are implemented as the same and share all model modules. During inference, to keep the process online, we only perform association between detections from the new-coming frame and existing trajectories. We realize this by iterating a sliding window with the stride of one frame.

**Training.** We train the association module by maximizing the likelihood of associating detections belonging to the same ground truth trajectory as expressed in Eq. 1. But Eq. 1 happens on one time step $t_i$ only. To speed up training, we calculate the association score on all $T$ frames of the sampled video clip at the same time and maximize the likelihood of the association aligned with the ground truths globally in the time window. The objective thus turns to

$$\prod_{q=t}^{t+T} P(\mathbf{M}^{\mathrm{asso}}_{j,\tau_q^j} = 1 | Q^{\mathrm{det}}, Q^{\mathrm{traj}}), \tag{2}$$

where $\tau_q^j$ is the ground truth index of the detection which should be associated with the $j$-th trajectory on the time step $q$. Therefore, by traversing the association of all trajectories, the training objective becomes the negative log-likelihood loss

$$L_{\mathrm{asso}} = -\sum_{j=1}^{M} \sum_{q=t}^{t+T} \log P(\mathbf{M}^{\mathrm{asso}}_{j,\tau_q^j} = 1 | Q^{\mathrm{det}}, Q^{\mathrm{traj}}). \tag{3}$$

On the other hand, trajectories can also be absent on some time steps because of occlusion or target disappearance. So similar to the practice of DETR (Carion et al., 2020) for detection and GTR (Zhou et al., 2022) for tracking, Eq. 3 has included the situation of associating a trajectory with "empty". Moreover, the main reason why mismatch happens is the features of objects of different identities being indiscriminative. Therefore, to encourage the representations from objects of different identities to be distinguishable, we design a feature discrimination loss in the form of triplet loss as

$$L_{\mathrm{feat}} = \max(0, \min_{u=1}^{N_P} ||\mathrm{Att}(f(F_{P_u}), f(F_B)) - f(F_B)||^2 - ||\mathrm{Att}(f(F_B), f(F_U^{bg})) - f(F_B)||^2 + \alpha), \tag{4}$$

where $f(\cdot)$ is the shared projection layers to project CNN features to feature vectors and $N_P$ is the number of part patches ($N_P = 4$ in our default setting). $\mathrm{Att}(\cdot, \cdot)$ is the operation of cross attention to generate attended features. $\alpha$ is the margin to control the distance between positive and negative pairs. $F_B$ and $F_{P_u}$ ($1 \le u \le N_P$) are the extracted features of the body area and the part sub-regions as explained already. $F_U^{bg}$ is the CNN features of the background area in the union box. We obtain the background features by setting the pixels of $B$ in the area of $U$ to be 0 and forward the processed patch of the union area into the feature encoder. We design Eq. 4 to encourage the projection network to pay more attention to the salient area on the body of target objects while less attention to the background area when processing the hierarchical part-whole representations. Also, it encourages the features of the background area in the union box, which probably belongs to another object target, to be distinguishable from the body features. This can be expected to decrease the chance of mismatch between neighboring objects. Finally, the training objective is

$$L = L_{\mathrm{asso}} + L_{\mathrm{feat}} + L_{\mathrm{det}}, \tag{5}$$

where $L_{\mathrm{det}}$ is an optional detection loss. In our default implementation, we finetune the detector at the same time when training the association modules.

**Inference.** We adopt the traditional sliding-window style to realize online influence. With a window size $T = 24$ and stride 1, we start from the first frame of the input video. On the first frame, every detection is initialized as an original trajectory. In each time window, we would generate

trajectories by detections within it. Then we use the association score in Eq. 1 to associate these trajectories with existing trajectories outside this time window. By averaging the detection-trajectory alongside detections of a trajectory, we get the trajectory-trajectory association scores, whose negative value serves as the entries in the cost matrix for the association assignment. And we adopt Hungarians matching to make sure the one-to-one mapping. Only when a pair of trajectories has the association score higher than a threshold $\beta = 0.3$, they are eligible to be associated. All unassociated detections on the new-coming frames will start new tracks.

## 4 EXPERIMENTS

### 4.1 EXPERIMENT SETUPS

**Datasets.** We conduct quantitative experiments on multiple multi-object tracking datasets, including MOT17 (Milan et al., 2016), MOT20 (Dendorfer et al., 2020) and DanceTrack (Sun et al., 2021). We focus on pedestrian tracking in this paper so pedestrian is the only category of objects of interest on all datasets. MOT17 and MOT20 are the classic and popular datasets in the area of pedestrian tracking but their scales are relatively small and have no official validation sets. DanceTrack, on the contrary, is a recently proposed dataset that is of a much larger scale and provides an official validation set with no overlap with the training set. DanceTrack focuses on the scenarios where targets are in the foreground so detection is not considered as the bottleneck as it is on MOT20. And Dance-Track mainly contains videos where targets have heavy occlusion, complex motion patterns, and similar appearances so it provides a good platform to study the robustness of the tracking algorithm.

**Evaluation Metrics.** The popular CLEAR evaluation protocol (Bernardin & Stiefelhagen, 2008) is based on single-frame-wise matching between the ground truth and predictions. This makes the metric emphasize single-frame detection quality rather than cross-frame association performance. MOTA, the main metric of CLEAR protocol, is also biased to the detection quality. To provide a more accurate sense of association performance in tracking, we mainly adopt the more recent HOTA (Luiten et al., 2021) metric set where the metric is calculated by the video-level association between ground truth and predictions. In the set of metrics, AssA emphasizes the association performance, and DetA stresses on the detection quality. HOTA is the main metric by taking both detection and association quality into consideration.

**Implementation.** We follow the common practice (Sun et al., 2020; Zeng et al., 2021; Cai et al., 2022) to use ResNet-50 (He et al., 2016) as the backbone network, which is pretrained on Crowdhuman (Shao et al., 2018) dataset first. Though advanced detector (Zhang et al., 2021a) is demonstrated as a key to boosting tracking performance, we want our contribution to be more from the improvement of the association stage. Therefore, on MOT17, we follow the practice of another transformer-based global association tracking method GTR (Zhou et al., 2022) to use the classic CenterNet Zhou et al. (2019; 2020) as the detector and all training details are aligned with it to make fair comparisons with this close baseline method. The CenterNet detector is pretrained together with the backbone on Crowdhuman to align with the common practice on this dataset. For the fine-tuning of association modules, we use a 1:1 mixture of MOT17 and Crowdhuman for MOT17. We fine-tune with only the MOT20 training set for evaluation on MOT20. For DanceTrack, we use its official training set as the only training set during finetuning. The image size is set to be $1280 \times 1280$ during training and the test size is 1560 for the longer edge during the test. During finetuning, the detector head is also fine-tuned as mentioned already. The training iterations are set to be 20k on MOT17/MOT20 and 80k on DanceTrack. We use BiFPN (Tan et al., 2020) for the feature upsampling. For the implementation of the transformer, we follow the practice of Zhou et al. (2022) to use a stack of two layers of "Linear + ReLU" as the projection layers and one-layer encoders and decoders. We use AdamW (Loshchilov & Hutter, 2017) optimizer for training whose base learning rate is set to be 5e-5. The length of the video clip is $T = 8$ for training and $T = 24$ for inference in a sliding window. We use $4 \times$ V100 GPUs as the default training device following some previous practice (Zhou et al., 2022; Zeng et al., 2021) but we will see that even using only one RTX 3090 GPU for training, our method can also achieve good performance. The training on MOT17 or MOT20 takes only 4 hours and the training on DanceTrack takes 11 hours.

### 4.2 BENCHMARK RESULTS

We benchmark our proposed method with existing methods now. The results on the MOT17-test dataset are shown in Table 1. HiPWA achieves the highest HOTA score among all transformer-

**Table 1:** Results on MOT17 test set with the private detections. **Bold** numbers indicate the overall best result and underlined numbers are the best transformer-based results.

| Tracker | HOTA↑ | AssA↑ | MOTA↑ | IDF1↑ | FP($10^4$)↓ | FN($10^4$)↓ | IDs↓ | Frag↓ |
|---|---|---|---|---|---|---|---|---|
| FairMOT (Zhang et al., 2021b) | 59.3 | 58.0 | 73.7 | 72.3 | 2.75 | 11.7 | 3,303 | 8,073 |
| Semi-TCL (Li et al., 2021) | 59.8 | 59.4 | 73.3 | 73.2 | 2.29 | 12.5 | 2,790 | 8,010 |
| CSTrack (Liang et al., 2020) | 59.3 | 57.9 | 74.9 | 72.6 | 2.38 | 11.4 | 3,567 | 7,668 |
| GRTU (Wang et al., 2021a) | 62.0 | **62.1** | 74.9 | 75.0 | 3.20 | 10.8 | 1,812 | **1,824** |
| QDTrack (Pang et al., 2021) | 53.9 | 52.7 | 68.7 | 66.3 | 2.66 | 14.7 | 3,378 | 8,091 |
| MAA (Stadler & Beyerer, 2022) | 62.0 | 60.2 | 79.4 | 75.9 | 3.73 | **7.77** | **1,452** | 2,202 |
| ReMOT (Yang et al., 2021) | 59.7 | 57.1 | 77.0 | 72.0 | 3.32 | 9.36 | 2,853 | 5,304 |
| PermaTr (Tokmakov et al., 2021) | 55.5 | 53.1 | 73.8 | 68.9 | 2.90 | 11.5 | 3,699 | 6,132 |
| TransMOT (Chu et al., 2021) | 61.7 | 59.9 | 76.7 | 75.1 | 3.62 | 9.32 | 2,346 | 7,719 |
| ByteTrack (Zhang et al., 2021a) | **63.1** | 62.0 | **80.3** | **77.3** | 2.55 | 8.37 | 2,196 | 2,277 |
| *Transformer-based Methods* | | | | | | | | |
| TransCt (Xu et al., 2021) | 54.5 | 49.7 | 73.2 | 62.2 | 2.31 | 12.4 | 4,614 | 9,519 |
| TransTrk (Sun et al., 2020) | 54.1 | 47.9 | 75.2 | 63.5 | 5.02 | 8.64 | 3,603 | 4,872 |
| MOTR (Zeng et al., 2021) | 57.2 | 55.8 | 71.9 | 68.4 | **2.11** | 13.6 | 2,115 | 3,897 |
| TrackFormer (Meinhardt et al., 2021) | - | - | 65.0 | 63.9 | 7.44 | 12.4 | 3,528 | - |
| GTR (Zhou et al., 2022) | 59.1 | 57.0 | 75.3 | 75.1 | 2.68 | 10.9 | 2,859 | - |
| MeMOT (Cai et al., 2022) | 56.9 | 55.2 | 72.5 | 69.0 | 3,72 | 11.5 | 2,724 | - |
| HiPWA (Ours) | 60.8 | 60.7 | 75.4 | 75.7 | 2,45 | 10,8 | 2,879 | 3,029 |

**Table 2:** Results on MOT20 test set with the private detections. **Bold** numbers indicate the overall best result and underlined numbers are the best transformer-based results.

| Tracker | HOTA↑ | AssA↑ | MOTA↑ | IDF1↑ | FP($10^4$)↓ | FN($10^4$)↓ | IDs↓ |
|---|---|---|---|---|---|---|---|
| FairMOT (Zhang et al., 2021b) | 54.6 | 54.7 | 61.8 | 67.3 | 10.3 | 8.89 | 5,243 |
| CSTrack (Liang et al., 2020) | 54.0 | 54.0 | 66.6 | 68.6 | 2.54 | 14.4 | 3,196 |
| GSDT (Wang et al., 2021b) | 53.6 | 52.7 | 67.1 | 67.5 | 3.19 | 13.5 | 3,131 |
| RelationT (Yu et al., 2021) | 56.5 | 55.8 | 67.2 | 70.5 | 6.11 | 10.5 | 4,243 |
| MAA (Stadler & Beyerer, 2022) | 57.3 | 55.1 | 73.9 | 71.2 | 2.49 | 10.9 | 1,331 |
| ByteTrack (Zhang et al., 2021a) | 61.3 | 59.6 | **77.8** | 75.2 | 2.62 | **8.76** | 1,223 |
| OC-SORT (Cao et al., 2022) | **62.1** | **62.0** | 75.5 | **75.9** | **1.80** | 10.8 | **913** |
| *Transformer-based Methods* | | | | | | | |
| TransCt (Xu et al., 2021) | 43.5 | 37.0 | 58.5 | 49.6 | 6.42 | 14.6 | 4,695 |
| TransTrk (Sun et al., 2020) | 48.5 | 45.2 | 65.0 | 59.4 | 2.72 | 15.0 | 3,608 |
| MeMOT (Cai et al., 2022) | 54.1 | 55.0 | 63.7 | 66.1 | 4,79 | 13.8 | 1,938 |
| HiPWA (Ours) | 53.0 | 51.1 | 65.8 | 64.4 | 3.64 | 13.7 | 3,948 |

based methods. But our method only achieves a comparable MOTA score with TransTrack (Sun et al., 2020) and GTR (Zhou et al., 2022), suggesting the superior part of HiPWA does not lie in the detection stage. The higher AssA score of our method also demonstrates its superior association performance.

MOT20 is a challenging dataset by containing scenes of crowded pedestrian flows. We report the results on the MOT20-test set in Table 2. Though HiPWA shows better performance than MeMOT (Cai et al., 2022) on MOT17, its performance is inferior on MOT20. This is probably related to the heavy and frequent occlusion on MOT20. It is common on MOT20 that a large portion of pedestrians' bodies is occluded for a long time. If the occlusion period is longer than the horizon of associating existing trajectories and new-coming detections, HiPWA will be likely to fail. On the other hand, the much longer temporal buffer of object appearance history maintained by MeMOT turns out more effective in such scenarios. However, we note that we design HiPWA with the main goal of demonstrating the hierarchical part-whole representation and choosing the most naive implementation for association heads to make it a computationally efficient baseline model. In the contrast, MeMOT requires 8×A100 GPUs for training to support the long-time history context buffering (22 frames v.s. 8 frames by HiPWA ) uses COCO (Lin et al., 2014) dataset as the additional pretraining data.

Next, we come to the benchmark on the DanceTrack dataset in Table 3. HiPWA achieves comparable performance with the best transformer-based methods. The association of HiPWA is inferior to MOTR (Zeng et al., 2021). MOTR has carefully designed global association and optimization modules. The global collective loss and query interaction module to propagate information frame by frame proposed by MOTR show good effectiveness. However, as a side-effect, its training and inference speed is much slower due to the heavy architecture. For example, training on MOT17 takes MOTR 2.5 days for MOTR on 8×V100 GPUs while only 4 hours on 4×V100 GPUs for our proposed method. And the inference speed is 6.3FPS for MOTR while 17.2FPS for our method on the same machine (V100 GPU). Compared to the close baseline GTR (Zhou et al., 2022), HiPWA

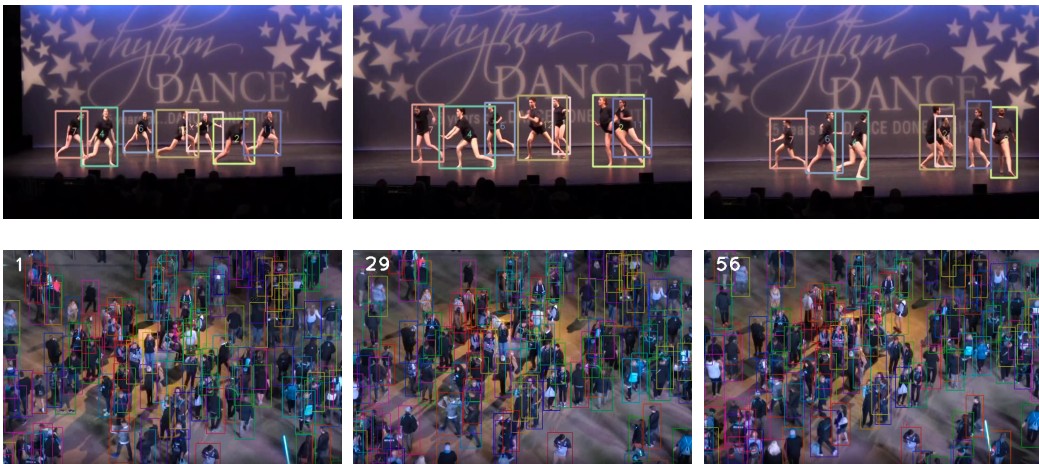

**Figure 2:** The upper line shows the results by HiPWA on three randomly sampled frames of a video in the DanceTrack-test set. The video is challenging to show camera motion, heavy occlusion, non-linear motion, and the crossover among targets at the same time. The bottom line shows results on a MOT20-test video where the pedestrians are in the crowd and heavily occluded.

achieves a more significant gap of outperforming on DanceTrack. Such an observation suggests our proposed part-whole hierarchical representation can be more powerful when the occlusion is heavy.

Given the results shown on the three benchmarks, we have demonstrated the effectiveness of our proposed HiPWA to be comparable to the state-of-the-art transformer-based multi-object tracking algorithms with a lightweight design. It builds a new baseline for future research in this line of works. The commonly adopted techniques of query propagation and iteration (Meinhardt et al., 2021; Sun et al., 2020; Zeng et al., 2021), deformable attention (Sun et al., 2020; Cai et al., 2022) and long-time feature buffering (Cai et al., 2022) are all compatible to be integrated with HiPWA .

**Table 3:** Results on DanceTrack test set. **Bold** numbers indicate the overall best result and underlined numbers are the best transformer-based results.

| Tracker | HOTA↑ | DetA↑ | AssA↑ | MOTA↑ | IDF1↑ |
|---|---|---|---|---|---|
| CenterTrack (Zhou et al., 2020) | 41.8 | 78.1 | 22.6 | 86.8 | 35.7 |
| FairMOT (Zhang et al., 2021b) | 39.7 | 66.7 | 23.8 | 82.2 | 40.8 |
| QDTrack (Pang et al., 2021) | 45.7 | 72.1 | 29.2 | 83.0 | 44.8 |
| TraDes (Wu et al., 2021) | 43.3 | 74.5 | 25.4 | 86.2 | 41.2 |
| ByteTrack (Zhang et al., 2021a) | 47.3 | 71.6 | 31.4 | 89.5 | 52.5 |
| OC-SORT (Cao et al., 2022) | **55.7** | **81.7** | **38.3** | **92.0** | **54.6** |
| Transformer-based Methods | | | | | |
| TransTrk(Sun et al., 2020) | 45.5 | 75.9 | 27.5 | 88.4 | 45.2 |
| MOTR (Zeng et al., 2021) | 54.2 | 73.5 | 40.2 | 79.7 | 51.5 |
| GTR (Zhou et al., 2022) | 48.0 | 72.5 | 31.9 | 84.7 | 50.3 |
| HiPWA (Ours) | 52.1 | 76.3 | 35.8 | 86.1 | 52.7 |

## 4.3 ABLATION STUDY

Though we provide the results on multiple benchmarks to show the efficiency and effectiveness of our proposed method, there are many variables in the design. We now ablate their contributions to the overall performance of HiPWA . Many previous works in the multi-object tracking community follow the practice of CenterTrack (Zhou et al., 2020) on MOT17 (Milan et al., 2016) to use the latter half of training video sequences as the validation set. However, this makes the ablation study on the validation set not always convincing because the data distribution of the training set and validation set is too close and the performance gap reflected on the validation set might shrink or even disappear on the test set. Therefore, we turn to DanceTrack (Sun et al., 2021) for the ablation study instead where an independent validation set is provided and of a much larger scale than previous MOT datasets.

In Table 4 and Table 5, we study the influence of video clip length in the training and inference stage respectively. The result suggests that training the association model with longer video clips

**Table 4:** Results of using different length of video clip during training.

| $T$ | HOTA↑ | DetA↑ | AssA↑ | MOTA↑ | IDF1↑ |
|---|---|---|---|---|---|
| 6 | 47.8 | 70.0 | 32.8 | 81.1 | 49.7 |
| 8 | 48.1 | 70.2 | 33.2 | 80.6 | 50.3 |
| 10 | 48.7 | 70.0 | 34.0 | 80.3 | 51.7 |
| 12 | 49.2 | 71.1 | 34.1 | 82.6 | 52.0 |

**Table 5:** Results of using different length of video clip during Inference.

| $T$ | HOTA↑ | DetA↑ | AssA↑ | MOTA↑ | IDF1↑ |
|---|---|---|---|---|---|
| 8 | 47.5 | 69.8 | 32.5 | 80.1 | 50.3 |
| 16 | 47.9 | 70.1 | 32.9 | 81.4 | 50.6 |
| 24 | 48.1 | 70.2 | 33.2 | 80.6 | 50.3 |
| 32 | 47.8 | 70.1 | 32.8 | 81.2 | 49.8 |

**Table 6:** Results on DanceTrack validation set to study the contribution from each level in our hierarchical representations to the association performance.

| | HOTA↑ | DetA↑ | AssA↑ | MOTA↑ | IDF1↑ |
|---|---|---|---|---|---|
| Body | 45.7 | 69.5 | 30.3 | 81.6 | 48.1 |
| Body + Part | 46.3 | 69.5 | 30.7 | 80.0 | 48.1 |
| Body + Union | 47.3 | 70.1 | 32.0 | 81.2 | 49.8 |
| Body + Part + Union | 48.1 | 70.2 | 33.2 | 80.6 | 50.3 |

**Table 7:** Results on DanceTrack validation set with different configurations for multiple training device choices. HiPWA shows good performance even given limited computation resources.

| Training Device | Train_len | Image Size | HOTA↑ | DetA↑ | AssA↑ | MOTA↑ | IDF1↑ |
|---|---|---|---|---|---|---|---|
| 1x RTX 3090-24GB | 6 | 1280 × 1280 | 47.8 | 70.0 | 32.8 | 81.1 | 49.7 |
| 1x V100-32GB | 8 | 1560 × 1560 | 48.0 | 70.8 | 32.6 | 82.4 | 50.1 |
| 4x V100-32GB | 8 | 1280 × 1280 | 48.1 | 70.2 | 33.2 | 80.6 | 50.3 |

can continuously improve performance. Limited by the GPU memory, we cannot increase the video clip length to longer than 12 frames here. On the contrary, during the inference stage, increasing the sliding window size does not significantly influence the tracking performance.

The hierarchical part-whole representation is the the main contribution of our proposed method. Considering that the hierarchical representation gathers information from three levels (Part, Body, Union), we study the contribution of each of them in Table 6. Compared to only using the features extracted from the bounding box (body) area, our hierarchical representation achieves a performance improvement of 2.4 points of HOTA and 2.9 points of AssA. On the challenging DanceTrack dataset, such improvement can be considered significant when they share the same detections. Also, integrating the features of the union area shows better effectiveness than solely integrating the features of body parts. This is probably because the cross attention between object body and union areas can provide critical information to compare object targets with their neighboring objects, which can prevent potential false association among them. On the other hand, the information about body parts is already contained in the object's body features. By concatenating the part features and body features, we can't introduce previously missing information pieces significantly.

Finally, as we aim to build a baseline model for future research in this area, we hope the proposed method is more accessible and computationally economic. We try different parameter configurations in Table 7. Even with only a single RTX 3090 GPU for training and inference, its performance is still quite close to our default setting which requires 4 × V100 GPUs. We hope this makes the notorious computation barrier of transformer-based methods not that terrible anymore.

## 5 CONCLUSION

In this paper, we propose to build discriminative hierarchical part-whole representations as the visual descriptor for objects in multi-object tracking. The representation is built upon only bounding box annotation and in three levels: Part, Body, and Union. They are designed to provide visual specificity of the object from the compositional, semantic, and contextual perspectives respectively. We further propose to use attention in the transformer to gather and process the visual features. The combination of these two aspects makes our method, namely Hierarchical Part-Whole Attention and HiPWA for short. The results on multiple datasets demonstrate its efficiency and effectiveness. We hope the study of this paper can provide new knowledge in the visual representation of objects and an advanced baseline model to solve multi-object tracking problems.

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
