# OpenReview forum: "Object Tracking by Hierarchical Part-Whole Attention"
_ICLR.cc/2023/Conference — Submitted to ICLR 2023_

### Official Review · Reviewer_rS4y · 2022-10-14

**Confidence:** 5
**Correctness:** 4
**Technical Novelty And Significance:** 2
**Empirical Novelty And Significance:** 2
**Recommendation:** 6

**Clarity, Quality, Novelty And Reproducibility:**

The overall structure is based on GTR but the context was not clear. The author should add add clear statement to acknowledge work by GTR paper. The proposed part-whole attention is well described. The discussion regarding L_feat should be expanded, especially alpha.

Experiments are conducted on different hardware setup and reported similar performance, which indicates certain level of reproducibility. But code is publicly available.

Modified a previous work to introduce more feature representation prior. Novelty is fair.

**Strength And Weaknesses:**

Transformer based tracking framework demonstrated promising performance. Research effort to improve those framework should be encouraged. This paper tried to explicitly guide the model to learn local object patches and surrounding object patches to improve performance. Experiments demonstrated the proposed part-whole attention improves tracking performance, both in the ablation study and on MOT17 and DanceTrack comparing to GTR.

- How was the structure in part-whole attention module designed? Did the author try other designs? Any insights?
- Given this paper is based on GTR, experiments should conduct on TAO dataset.


**Summary Of The Paper:**

This paper proposed modification for multiple object tracking framework based on GTR[Zhou et.al 2022]. Instead of use the feature representation from detection bounding boxes. This paper proposed to use split bounding boxes and surrounding image patch if there are other objects located close to the current object, which aims to tackle occlusion and avoid identity switch cases.

The proposed methods performs comparable with state-of-the-art methods on MOT17, MOT20 and DanckTrack benchmark.

**Summary Of The Review:**

This paper proposed Part-Whole Attention module for GTR framework for multiple object tracking task. The proposed module are clearly described and experiments demonstrated the effectiveness.

More experiments on TAO dataset should be included to compare with its baseline method GTR. Discussions of the structure design for Part-Whole Attention module should be added

---

> ### Author Response · Authors · 2022-11-10
> **Response to Reviewer rS4y**
>
> We appreciate your review and recognizing its contributions. Here are our responses to your questions.
>
> **Q1: Connection with GTR[1]**.
>
> **A1:** The implementation of this work is before the publication of GTR. When we came up with the idea of hierarchical representation, to have enough detections to compare in a batch, we follow an earlier work VisTR[2] to associate objects in a video clip. Such a scheme is also called "global association" by GTR. To keep our implementation clean to demonstrate the effectiveness of the part-whole representation, we did not adopt the learnable query and deformable convolution but kept the positional encoding. Later, we found the release of GTR and we align the settings of our implementations with it on MOT17 as it becomes the closest baseline and we have to make the comparison fair.
>
> **Q2: Insight and intuition behind the part-whole attention module**.
>
> **A2:** The design of the part-whole attention module is inspired by some observations:
>
> 1. Tracking a part of an object's body instead of its full body can be more effective, especially in occlusion.
>
> 2. Failures of tracking frequently happen due to mismatching neighboring objects. Observing the objects with the neighboring objects can provide a context to help distinguish between them.
>
> Inspired by the observations, we thought that we should also take the levels of object body patches and the union of neighboring objects into appearance matching during association. And the attention mechanism is a good tool to fuse the features. Motivated by such insight, we design the part-whole attention.
>
> **Q3: Design choice of the part-whole attention module**.
>
> **A3:** The proposed part-whole attention module is illustrated in Figure 1 (right) and Section 3.2, we recap some design choices here:
>
> * The "Feature Encoder" is the Resnet-50 network as explained in Page 6.
>
> * We use RoI align to extract features from the area of interest of different levels.
>
> * We use a two-layer projection network to process the extracted features.
>
> * After the cross-attention, we have another MLP network to project the features to the final tokens.
>
> There can be some variations in the design choice, including:
>
> 1. the number of patches: As the GPUs with the largest memory we can access are V100-32GB and increasing the patch number to more than 2x2, e.g. 3x3, requires more than 32GB GPU memory. We thus can only choose 2x2 patches as in the paper. The ablation study in Table 6 has suggested the effectiveness of using part (patch) features but more ablation studies can be helpful.
>
> 2. The projection layers between the feature encoder and cross-attention module: without using the projection layers, the features for cross-attention are the original CNN features. Because the features, though of the same size, come from different resolutions, their CNN features can't be aligned effectively. So we use a projection layer to align their representations. This layer is similar to the projection layers in common transformer usage when generating tokens.
>
> 3. We use concatenation for Part-Body features but cross-attention for Union features. This is motivated by the insight that cross-attention is used to discover consistent areas on object bodies. And the union area can provide contextual cues to avoid mismatch. Because the distinguishable area can come from both the Part level and the Body part, they should be leveraged equally here so we concatenate them.
>
> **Q4: experiments on TAO dataset**
>
> **A4:** Our focus is to demonstrate the effectiveness of the proposed hierarchical representation and we want to decrease the influence of other noise. For tracking on TAO, the prediction of object categories is extremely inaccurate and causes a big noise when justifying whether the hierarchical representation is helpful as, by default, we only compare features of objects of the same category. Therefore, we don't include TAO and only evaluate on the datasets with no problem of object classification.
>
> **Q5: discussion about $L_{feat}$**
>
> **A5:** $L_{feat}$ is the classic and widely adopted triplet loss and $\alpha$ is known as the “margin” to control the allowed distance between positive and negative pairs. The positive pairs are the features of the object's body area and the attended most distinguishable part features. The negative pairs are the features of the object's body area and the background area. We want to encourage the model to attend to the distinguishable part of the object body while making the features from the background and the object body less similar. This loss helps to consistently match the same object across frames while avoiding mismatching it with the “background” area in the union box which contains a neighboring object instance. We follow the default implementation and parameter by Pytorch for this loss (torch.nn.TripletMarginLoss).
>
> Reference:
>
> [1] “Global Tracking Transformers”
>
> [2] "VisTR: End-to-End Video Instance Segmentation with Transformers"

---

### Official Review · Reviewer_YxoG · 2022-10-26

**Confidence:** 3
**Correctness:** 3
**Technical Novelty And Significance:** 2
**Empirical Novelty And Significance:** 2
**Recommendation:** 6

**Clarity, Quality, Novelty And Reproducibility:**

The combination of 3 levels of representation especially the union of overlap using attention is somewhat creative.

**Strength And Weaknesses:**

The strength of the paper is the fusing of the three levels of representation for multi-object tracking associations. The weaknesses of the paper are some of the notations and areas are not clearly presented.
1. The notations of M and N in section 3.1 are not defined when appearing in the first and second part. This causes confusion when reading the paper.
2. N should be T in 1<=i<=N in the third page.
3. In page 4, "we can tract the CNN features from them", what is the CNN architect to extract the features?
4. In page 6 Inference section, "whose negative value serves as the entries in the cost matrix". The cost value should not be some negative values.

Another weakness of the approach is the lack of motion feature in the association. In the association, only appearance features are considered. This can lead to association errors when two objects have similar appearances (similar color of clothes) even they are farther apart in the space. In the Detection and Feature Extraction section, detection features are extracted from a video clip of T frames and then the detection from each one of the T frames is associated with the trajectories. There can be significant motions in the T frames and appearance would be hard to handle the discriminability of the objects.

**Summary Of The Paper:**

The paper proposes a hierarchical representation of objects for multi-object tracking. The hierarchical representation consists of 3 levels: part, whole and union of overlapped objects. The proposed approach demonstrated good performance on multiple pedestrian/dance public datasets.

**Summary Of The Review:**

Overall, this work shows somewhat creativity, but only considers the appearance features can be insufficient to solve challenging tracking cases.

---

> ### Author Response · Authors · 2022-11-10
> **Response to Reviewer YxoG**
>
> We appreciate your review of this work and recognizing its contributions. Please find our responses to your questions below.
>
> **Q1: some notations are not clear**
>
> **A1:** We have updated the draft with more notation definitions in that section, the revised content is highlighted in blue. And we clarify some points here:
>
> 1. In section 3.1, $N$ is the number of detections on the $T$ frames. $M$ is the number of existing trajectories.
>
> 2. $1 \leq i \leq N$ is the correct notation. $t_i$ is the time step for the $i$-th detection. Given there are in total $N$ detections, the range of $i$ should be $[1,N]$.
>
> 3. In paragraph 4, the CNN architect to extract the features is related to the practical implementation. For our default implementation, as explained in the "Implementation" paragraph at Page 6, we use ResNet-50 to extract the CNN features.
>
> 4. The absolute value of entries in the cost matrix does not matter because the association cares only about the relative values of different entries in the cost matrix. By default, entries with a lower cost should be given higher priority. As we want to select the pairs with higher association scores, we use the negative values of the association scores in the cost matrix to align with the common practice.
>
> **Q2: The method does not use motion information**
>
> **A2:** Yes, appearance and motion are the two important cues in associating targets. However, the focus of this paper is to demonstrate the effectiveness and efficiency of the proposed hierarchical part-whole representation in tracking, we evaluate our idea with a minimum design to avoid the influence from other components when estimating the effectiveness of the proposed representation. Further adding more information, of course including motion, can potentially boost the performance but that is out of the focus of this work. Also, existing tracking algorithms are mostly appearance-matching-based methods[1,2,3,4,5]. We expect more advanced methods combining both motion and appearance cues in multi-object tracking but adding motion cues would make it harder to determine whether the performance gain purely comes from that the proposed hierarchical representation can represent object visual occupancy in a more distinguishable way.
>
> Reference:
>
> [1] “QDTrack: Quasi-Dense Similarity Learning for Appearance-Only Multiple Object Tracking“
>
> [2] “TransTrack: Multiple Object Tracking with Transformer”
>
> [3] “Global Tracking Transformers”
>
> [4] “TrackFormer: Multi-Object Tracking with Transformers”
>
> [5] “FairMOT: On the Fairness of Detection and Re-Identification in Multiple Object Tracking”

---

### Official Review · Reviewer_hWAz · 2022-10-28

**Confidence:** 3
**Correctness:** 4
**Technical Novelty And Significance:** 3
**Empirical Novelty And Significance:** 3
**Recommendation:** 8

**Clarity, Quality, Novelty And Reproducibility:**

this paper is clearly written and the information is enough to reproduce the experiments.

**Strength And Weaknesses:**

strength:
1. the idea of Hierarchical Part-Whole representation of target object in MOT seems interesting;
2. the combination of Hierarchical Part-Whole Attention and Transformer works well on existing benchmark datasets;
3. the paper is easy to follow and understand.


weakness:
1. the source code will be released or not is unclear. it is an interesting idea, but the implementation is also complicated. If the code is not available, maybe it is hard for other researchers to follow.
2. the running efficiency is not real-time. i.e., not fast enough for practical applications.

**Summary Of The Paper:**

This paper proposes an interesting 'Hierarchical Part-Whole Attention' for multi-object tracking. The proposed module is integrated with transformer network and achieves good performance (comparable or even better results than SOTA mot trackers). The overall training efficiency is also good, i.e., 4 hours on 4*v100 GPUs, while other Transformer based trackers may need days. This paper is well-written and organized, and I believe it will be a good baseline for future works to compare and in-depth development on this framework.

**Summary Of The Review:**

the idea is reasonable,
the framework is not very complicated for MOT.
the training time is acceptable, but the inference is fast;
this paper is well-written and easy to follow.

---

> ### Author Response · Authors · 2022-11-10
> **Response to Reviewer hWAz**
>
> We appreciate your review of this work and recognizing its contributions. Here we respond to your questions:
>
> **Q1: The concern about reproducibility**
>
> **A1:** When implementing the network, we choose a minimum design on purpose that we remove unnecessary components that can potentially introduce noise when comparing our proposed hierarchical part-whole representation with the traditional bounding-box-based representation. Thus we don’t use some modules such as positional encoding, learnable object query, etc in the implementation. We hope such a design helps following researchers to build their models upon it more easily and make the contributions of the proposed representation more clear. Also, the source code of this work would be released to the public upon acceptance.
>
> **Q2: The inference of the proposed method is not real-time.**
>
> **A2:** Yes, our proposed method shares the shortcoming of the time efficiency of all other transformer-based methods. We noticed and discussed it in Sections 4.2 and 4.3. Thanks to the global association paradigm and the minimum design we choose, as far as we know, our proposed method already achieves one of the best time efficiency among the transformer-based methods we investigated. But there is still a gap to enable the real-time running of a transformer-based method to do multi-object tracking on existing hardware.

---

> > ### Comment · Reviewer_hWAz · 2022-11-19
> > **I like this paper and current version can be accepted for publication.**
> >
> > Thanks for the feedback. I like the idea of this work and believe it will be beneficial for the MOT community.

---

### Author Response · Authors · 2022-11-10
**General Response to All Reviewers**

We thank all the comments and suggestions from the reviewers. We have revised the paper draft and will keep polishing it. The updated content is highlighted in blue. Here, we summarize the motivation and intuition behind this work again.

**Motivation and intuition of this work:** In the paper, we are motivated by the observations that when the occlusion is heavy in the video, focusing on a part instead of the full body can be more effective to track an object over time. On the other hand, many failures in multi-object tracking happen by mismatching neighboring objects. However, many widely used appearance matching-based tracking algorithms only compare the appearance features extracted from the whole object bounding box area. We thus believe that we could also use a “part” or the contrast between an object with its neighbors to define its visual occupancy.  Then, we provide a transformer-based implementation to demonstrate the effectiveness of such a hierarchical representation in the task of multi-object tracking.

---

### Decision · Program_Chairs · 2023-01-20

**Decision:**

Reject

**Justification For Why Not Higher Score:**

The technical novelty is very incremental, and very specific to MOT. The experiment results are mixed. The method is not fully tested against its closest baseline.  Some interesting ablation study is missing.

**Justification For Why Not Lower Score:**

n/a

**Metareview: Summary, Strengths And Weaknesses:**

**Summary:** The paper proposes a hierarchical representation of objects for multi-object tracking. The hierarchical representation consists of 3 levels: part, whole and union of overlapped objects. The proposed approach demonstrated good performance on multiple pedestrian/dance public datasets.

**Strengths:** Promising results on the tested benchmarks.  The paper is easy to follow.

**Weaknesses:** The following weaknesses were identified:

1. running speed is not real-time, thus not practical. [hWAz]
2. notations could be improved. [YxoG]
3. no motion features are used in association, which can cause errors when two objects have similar appearances [YxoG]
4. what is the design principle for the part-whole attention module? [rS4y]
5. Need experiments on TAO dataset to compare with GTR [rS4y]
6. more discussion on Lfeat [rS4y]

**Discussion:**
The authors wrote a response to address the concerns, and the reviewers maintained their positive ratings (668).
However, the AC noted that the technical/empirical ratings from Reviewers rS4y and YxoG:
- Technical Novelty And Significance: 2: The contributions are only marginally significant or novel.
- Empirical Novelty And Significance: 2: The contributions are only marginally significant or novel.

After reading the paper, the AC agreed with the two reviewers that the novelty and technical contribution is quite low -- the main contribution is the particular way of performing cross-attention between object patches, whole object, and context to form a token representation of the object. However, the paper provides no analysis of the representation to show its suitability (e.g., visualizations of when it works and how it works), and the proposed representation is highly specific to MOT.  The rest of the MOT architecture using transformers follows the global association paradigm as in GTR (Zhou et al 2022) and VisTR (Wang et al 2021). The empirical results are also marginally significant -- the proposed method outperforms other transformer-based methods on 1 of 3 tested datasets (in terms of HOTA), although the proposed work is faster. However, there is not really an in-depth analysis of the speed-accuracy tradeoff to illustrate this. Given the similarity with GTR, the proposed method should have been compared on the TAO dataset (as suggested by rS4y), which GTR tests on. Finally, the proposed work includes the triplet loss (Eq 4), which is different from GTR, and there was no ablation study to show the effect of the extra loss term. Thus the real effect of the proposed representation is not well understood.

Given the above, the AC found that the "6" ratings (marginally above acceptance threshold) from Reviewers rS4y and YxoG were inconsistent with the technical/empirical ratings.  When asked about this during the discussion, no reviewers offered to champion the paper. Therefore, the AC decided that "6" ratings were not appropriate, and thus decided to reject the paper.


**Summary Of Ac-Reviewer Meeting:**

n/a